# Effects of a Combination of Polynucleotide and Hyaluronic Acid for Treating Osteoarthritis

**DOI:** 10.3390/ijms25031714

**Published:** 2024-01-30

**Authors:** Seung Hee Choi, Hyun Chul Kim, Seul Gi Jang, Yeon Jae Lee, Jun Young Heo, Gi Ryang Kweon, Min Jeong Ryu

**Affiliations:** 1Joonghun Pharmaceutical Co., Ltd., 15 Gukhoe-daero 62-gil, Yeongdeungpo-gu, Seoul 07236, Republic of Korea; shchoi@joonghun.com (S.H.C.); hckim@joonghun.com (H.C.K.); sgjang@joonghun.com (S.G.J.); yjlee@joonghun.com (Y.J.L.); 2Department of Biochemistry, College of Medicine, Chungnam National University School of Medicine, Daejeon 35015, Republic of Korea; jyheo@cnu.ac.kr (J.Y.H.); mitochondria@cnu.ac.kr (G.R.K.)

**Keywords:** polynucleotide, hyaluronic acid, knee OA, anterior cruciate ligament transection, medial meniscectomy

## Abstract

Knee osteoarthritis (OA), an age-related degenerative disease characterized by severe pain and disability, is treated using polynucleotides (PNs) and hyaluronic acid (HA). The intra-articular (IA) injection of HA has been studied extensively in both animal models and in humans; however, the efficacy and mechanisms of action remain unclear. In addition, there has been a paucity of research regarding the use of PN alone or in combination with HA in OA. To investigate the effect of the combined injection of PN and HA in vivo, pathological and behavioral changes were assessed in an OA model. Anterior cruciate ligament transection and medial meniscectomy were performed in Sprague-Dawley rats to create the OA animal model. The locomotor activity improved following PNHA injection, while the OARSI grade improved in the medial tibia and femur. In mild OA, TNFα levels decreased histologically in the PN, HA, and PNHA groups but only the PNHA group showed behavioral improvement in terms of distance. In conclusion, PNHA exhibited anti-inflammatory effects during OA progression and improved locomotor activity regardless of the OARSI grade.

## 1. Introduction

Historically, osteoarthritis (OA) has been considered an age-related degenerative disease characterized by cartilage degeneration, subchondral bone changes, meniscus degeneration, and osteophyte formation due to mechanical wear [1,2]. However, the more important role of crosstalk between various joint structures and local inflammation is now widely recognized [3,4]. The infrapatellar fat pad (IFP) is fatty tissue beneath the patella, and inflammation of this tissue is known to affect the progression of OA [5,6]. In addition, synovial fibrosis and fibrocartilage production also occur in OA joints, and they therefore play important roles in the development and progression of the disease [7]. Although OA is caused by mechanical friction stress on the joints, the disease progresses through several orchestrated processes, including inflammation and fibrosis, causing pain, swelling, and deformity of the joints [8].

Disrupted cytokine balance activates catabolic enzymes, such as matrix metalloproteinases (MMPs) and a disintegrin-like and metalloproteinase with thrombospondin motif (ADAMTs) [9,10]. Matrix degeneration induced by these enzymes leads to damage to the cartilage and other structures within the joint, altering the tissue biomechanics [11]. The most important inflammatory mediators in the pathogenesis of OA are tumor necrosis factor (TNF)-α, interleukin (IL)-1β, and IL-6, which are activators of various signaling pathways that activate other proinflammatory cytokines and MMPs and pathological processes [9,12,13,14,15]. TNF-α and Ils activate multiple signaling pathways to increase the expression of cyclooxygenase-2 (COX-2), and consequently prostaglandin E2 (PGE-2), which subsequently affects cartilage degradation and osteophyte formation [16,17].

Several pharmacological and non-pharmacological treatment options are available for patients with symptomatic knee OA. Nonsteroidal anti-inflammatory drugs (NSAIDs) serve as the first-line treatment to alleviate pain. However, chronic NSAID use often results in persistent pain and disability, with more than half of such patients discontinuing them within a year [18,19,20,21]. In addition, NSAID use has been reported to be associated with three-, four-, and two-fold increases in the risks of stroke, cardiovascular death, and death from all causes compared to placebo, respectively [19,22,23]. It is therefore recommended that NSAIDs be used intermittently or periodically and to avoid continuous long-term use.

For patients with an inadequate response to pharmacological treatment or comorbidities limiting medical options, IA injection therapy, particularly with hyaluronic acid (HA), plays a crucial role. HA has a non-sulfated glycosaminoglycan structure composed of D-glucuronic acid and N-acetylglucosamine units. HA is naturally present in various tissues, including the skin, eyes, thoracic lymph, synovial fluid, and cartilage [24,25]. It is involved in a number of biological processes, including embryonic development, wound healing, cancer progression, angiogenesis, inflammation, and bone regeneration [26,27,28]. HA-based biomedical products have been developed to target various diseases due to their biocompatibility, biodegradability, nontoxicity, and nonimmunogenicity [29]. In OA disease, HA is the most common IA agent, functioning as a visco-supplement to reduce knee OA-associated pain by enhancing joint function under physical stress. As the synovial fluid in OA joints has a lower HA concentration than in healthy joints [30], IA injection of HA restores the viscoelasticity of the synovial fluid in OA patients, reducing the stress or absorbing the load on the joint. Furthermore, HA exhibits anti-apoptotic, anti-inflammatory, anti-angiogenic, and anti-fibrotic properties [24]. HA is further classified into high (≥3000 kDa), medium (1500–3000 kDa), and low (≤1500 kDa) molecular weight (MW) groups [31]. High MW-HA has been reported to have superior chondroprotective, anti-inflammatory, rheological, analgesic, mechanical, and proteoglycan production properties [32]. However, despite numerous pre-clinical and clinical reports on its efficacy and safety, the effectiveness of HA in knee OA remains controversial.

In this context, Condrotide (20 mg/mL, Mastellisa, Italy), an innovative medical product composed of polynucleotides (PN), has been developed and utilized since 2010 [33,34,35,36,37]. PNs, derived from the milt of salmon or trout, consist of (deoxy)ribonucleotides containing purines and pyrimidines. Being highly viscoelastic because of its large molecular weight and water-binding capacity, PN functions through both non-pharmacological and pharmacological mechanisms, including collagen production, tissue regeneration, and inflammation reduction via adenosine A (2A) receptor stimulation [38,39,40]. Several clinical studies have demonstrated that PN products yield functional improvements in knee OA, providing enhanced pain control compared to IA injections of HA [34,35]. Consequently, IA injections of PN have been proposed and utilized as an alternative to HA for visco-supplementation over the past decade [41,42].

Considering the anti-inflammatory, tissue regenerative, and visco-supplementation effects of both PN and HA, there is an expectation that their combination may offer superior benefits for OA patients experiencing persistent pain and worsening disease. To date, two clinical studies have been reported comparing PNHA combinations with HA alone. Both the PNHA and HA groups showed significant improvements in the total Knee Society Score (KSS) and pain items, and long-term reductions in the Western Ontario and McMaster Universities (WOMAC) scores. Moreover, the PNHA group showed better results than the HA group in the KSS and WOMAC scores [41,43]. However, these clinical studies focused only on pain and lacked information on the pathological characteristics and anti-inflammatory properties, and there have been no studies to date on the combined effects of PN and HA in animal models of OA.

Anterior cruciate ligament transection (ACLT) and medial meniscectomy (MMx) induced rat OA models exhibit several characteristics seen in human and other animal OA models, including progressive articular cartilage degradation, subchondral bone sclerosis, and osteophyte formation [44]. Models with only MMx or ACLT alone demonstrated very mild pathological findings of OA after 8 weeks. Therefore, we opted for a combined ACLT and MMx model for evaluating the PNHA combination.

To assess the efficacy and mechanism of the PNHA mixture in relation to OA severity, we conducted investigations into the effectiveness of PN alone and PNHA in ACLT and MMx models.

## 2. Results

### 2.1. IA injection of PNHA/a Delays Cartilage Degeneration

The current concentration of the raw material for IA injection, utilizing PN as the main ingredient in orthopedics, is 20 mg/mL (2%). As HA products of various concentrations are used in orthopedics [45], the lowest concentration that showed effectiveness in an OA animal model was selected [46]. To evaluate the efficacy of the PNHA complex, 20 mg/mL (2%) PN and 5 mg/mL (0.5%) HA were mixed (PNHA/a) and then injected into an OA animal model. Given the high viscoelastic properties of both PN and HA, the PNHA/a formulations were uniformly mixed using physicochemical methods and packaged in pre-filled syringes. During the PNHA/a manufacturing, a constant-temperature heating process was employed, revealing an elastic modulus of 178.4667 Pa for PN and 141.625 Pa for PNHA/a, slightly lower than that of PN alone (Figure 1B).

An animal OA model was established through combined ACLT and MMx surgery on the hind limbs of 10-week-old Sprague-Dawley rats [44]. To assess the PNHA/a efficacy, IA injections were commenced after 2 weeks of postsurgical stabilization. For the PN IA injections, performed five times due to its five-time dosage schedule, the open-field locomotor activity test (OFT) was conducted at the 8th week. Subsequently, hind limb joints were extracted for pathological examination (Figure 1A).

Safranin O-Fast Green Staining for intra-cartilage proteoglycan evaluation was used to confirm articular cartilage damage and proteoglycan loss in the ACLT MMx model. The IA injection of PNHA/a rescued articular cartilage damage and proteoglycan loss, with the PN group showing a trend similar to that of the PNHA/a group (Figure 1C). Furthermore, the Osteoarthritis Research Society International (OARSI) score [47], which assesses the degree of articular cartilage destruction, was used to confirm the effect of PNHA/a. In the case of the ACLT MMx model, the OARSI scores of the medial femur and tibia increased significantly following the removal of the medial meniscus, but there was no significant difference in the lateral region. In the medial tibia, OARSI decreased significantly from stage 3.5 in the OA group to stage 2.7 in the PNHA/a and PN groups (Figure 1D), and that in the medial femur decreased from stage 2.2 to stage 1.6 in the PN group (Figure 1E). The grade of the medial tibia also decreased from grade 5 to grade 4 in the PNHA/a group and PN group (Figure 1D,E), with no improvement observed in the medial femur (Figure 1E). Calculating the OARSI score by combining the stage and grade, the medial tibia area in both the PNHA/a and PN groups exhibited approximately 40% improvement compared to the OA group, with no significant differences between the two groups (Figure 1D).

### 2.2. PNHA/a Rescues Locomotor Activity in the OA Rat Model

OFT was performed to measure disturbances in the distance and speed traveled due to pain occurring in OA [48,49]. Following OFT, the OA group exhibited an approximately 10% decrease in moving distance and speed compared to the sham group (Figure 2A,B). Both the PNHA/a and PN groups showed recovery to about 90% compared to the OA group. No significant differences were observed between the PNHA/a and PN groups, consistent with the OARSI score findings.

These findings indicate that in the ACLT MMx-induced OA animal model, PN mixed with 0.5% HA improved OA both pathologically and functionally. However, there was no additional effect attributable to the HA mixture.

### 2.3. IA Injection of PNHA/b Suppresses MMP13 Expression

To assess the significance of the anti-inflammatory effect of PN and HA raw materials on exercise activity in OA, the commercial products Cureran (2% PN) and Lufla (1% HA) were used as controls. Furthermore, to explore whether the combination of these two raw materials synergistically influenced the anti-inflammatory and exercise activity in OA, PNHA/b was prepared by mixing 2% PN and 1% HA. Modifying the product mixing conditions and increasing the HA concentration improved the modulus of elasticity to 235.6 Pa compared to PNHA/a (Figure 1B and Figure 3B).

In the OARSI level 4 cases, the product’s efficacy could not be confirmed due to severe articular cartilage damage. To evaluate inflammation in the OA animal model, surgery was performed on 8-week-old rats to reduce the load on the knee, subsequently lowering the OARSI grade (Figure 3A). The combination of hematoxylin and eosin (H&E) staining and safranin-O Fast Green staining revealed an OARSI grade 2 in the OA model, with no significant improvement in the Cureran, Lufla, and PNHA/b groups (Appendix A).

To confirm pathological changes in the tibia and femur, immunostaining for MMP13 and COLII was performed. In the OA model, decreased COLII expression and MMP13 induction were observed in both the tibia and femur (Figure 3C,D). However, COLII expression showed no improvement in any group (Figure 3C), and MMP13 decreased only in the HA-containing group, PNHA/b group, and Lufla group (Figure 3D,E).

### 2.4. IA Injection of PNHA/b Reduces TNFα Levels in OA Cartilage

Immunofluorescence staining to assess inflammatory cytokines, including TNFα, IL6, and IL1β, revealed increased TNFα (Figure 4A,C) and IL6 (Figure 4B,D) levels in the medial tibia region of an OA animal model, while IL1β showed a slight decrease (Appendix A). PNHA/b, Cureran, and Lufla demonstrated a reduction in TNFα levels, with no observable effect on IL6. When the staining intensity in the medial tibial region was quantified, it was found that despite the trend that PNHA/b may be more effective in reducing TNFα expression than Cureran and Lufla, the difference did not reach statistical significance (Figure 4C).

Consequently, both PN and HA individually exhibited anti-inflammatory effects by suppressing TNFα, but there was no synergistic effect of the two raw materials.

### 2.5. PNHA/b Rescued Locomotor Activity in the OA Rat Model

Unlike the grade-4 OA model (Figure 2), the mild grade-2 OA model exhibited a decrease in the moving distance, while the velocity remained comparable to that in the sham controls (Figure 5A,B). PNHA/b, Cureran, and Lufla demonstrated anti-inflammatory effects by reducing the amount of TNFα. However, when the locomotor activity was assessed through the OFT, the moving distance significantly improved only in the PNHA/b group, reaching a level comparable to that of the sham group (Figure 5A).

Based on these results, the anti-inflammatory effect of TNFα inhibition may not be directly related to exercise activity in ACLT MMx-induced OA. In conclusion, PNHA/b exhibited anti-inflammatory effects during OA progression and ultimately enhanced locomotor activity by improving the OA stage and grade.

## 3. Discussion

OA is a disease that causes pain, swelling, and joint deformation through various systematic processes, including mechanical friction stress, inflammation, and joint fibrosis. Oral NSAIDs are prescribed in the early stages of OA, but due to several side effects, IA of HA and PN are recommended as alternative treatments.

Many clinical and animal tests have been conducted on HA based on the concentration and size of HA raw materials. However, in the case of PN, there has been only one previous report in which PN was administered orally in a monoiodoacetate (MIA) animal model of OA [50], so the effectiveness of IA injection was not evaluated. The effectiveness of PN and HA mixtures has only previously been evaluated in clinical trials of Condrotide Plus (1% PN with 1% Medium-MW HA); the results showed differences according to the ratios of PN and HA [15,17]. Furthermore, both Condrotide Plus and PN were evaluated for efficacy compared to HA products in clinical studies.

In a clinical study that compared the efficacy and safety of the IA injection of a mixture of PN and HA with the IA injection of HA alone, the combination was found to improve knee function and joint pain more effectively than HA alone in knee OA patients [16,18]. However, the effectiveness of PN and HA mixtures has only been evaluated in clinical trials of Condrotide Plus, and differences according to the ratios of PN and HA have not been examined. Both Condrotide Plus and PN products have been evaluated for efficacy in comparison to HA products in clinical studies.

In our study, using the same MMW HA but with a commercially available 2% PN product (Cureran) as a control, we doubled the concentration of PN to 2%. Regardless of the HA concentration, the OFT results demonstrated that the mixture of PN and HA significantly improved the locomotor activity compared to Lufla, consistent with the results of previous clinical trials. However, the OFT results for PNHA/a (0.5% HA) and PNHA/b (1% HA) showed no difference in the improvement in exercise activity based on HA concentration.

In the animal experiment, commercially available Cureran and Lufla were used as controls for clinical comparison. However, from a scientific perspective, using PN alone, manufactured from the same raw material, is deemed more appropriate as a control group. The 2% PN we produced exhibited similar improvements in locomotor activity as the IA injection of the PNHA mixture. These results suggest that the combination of PN and HA does not have a synergistic or additive effect on behavioral outcomes in animal OA models compared to PN alone. When comparing PN-only products, the 2% PN we manufactured showed enhanced locomotor activity, while the commercially available product Cureran did not exhibit any improvement in locomotor activity. The physicochemical properties of the raw materials in our PN-only preparation and the commercially available Cureran, despite having the same composition, were expected to differ. The viscoelasticity of the two products varied by approximately 100 Pa, attributable to differences in the nature of raw materials and the manufacturing process. Although both PN products contain identical DNA extracted from salmon milt, further investigation is needed to understand the impact of these differences on locomotor activity.

Common behavioral assessments for rodent OA models include gait analysis, running wheel tests, rotorod assessments, weight bearing evaluations, and OFT, among others [20]. The OFT is particularly useful for simultaneously measuring different behaviors in rodents, such as exploratory behavior, gross motor activity level, and anxiety. Reduced motor activity is a common consequence of OA pain, but when and how to measure changes in motor activity in surgical OA models remain unclear. Sambamurthy et al. compared animals with a destabilized medial meniscus to unoperated controls and reported reduced locomotor activity, as assessed via OFT at 8 weeks [51,52]. Meanwhile, other studies reported that there were no differences in the OFT results compared to the sham control in the cruciate ligament transection model [53] or destabilization of the medial meniscus model [54]. Previous studies have also demonstrated that mice treated with MIA exhibit a significant decrease in locomotor activity compared to untreated controls, suggesting the role of joint inflammation in reduced activity [55,56]. Our ACLT MMx model also demonstrated cartilage inflammation, with a reduction of more than 10% in the OFT in the OA model. Thus, despite the milder inflammatory state compared to other models, the OFT can serve as an indicator of pain-associated loss of motor activity in the ACLT MMx model.

A previous study demonstrated that PN treatment in an OA cell model is associated with an anti-inflammatory response [57]. Additionally, PN treatment in human chondrocyte cells (CHON-001) reduced inflammatory genes, and oral administration of PN in the MIA model suppressed the inflammatory gene levels in joints [50]. Consistent with these findings, the TNFα protein levels decreased in the joints in all of the IA injection groups (PNHA, Cureran, and Lufla). While all of the groups exhibited reductions in TNFα, only PNHA improved locomotor activity, suggesting that the viscoelasticity of the injection is more crucial for pain-related locomotor activity than the anti-inflammatory effect of the raw material.

This was the first study to confirm the effects of the injection of PN into the joint space of OA animal models and to verify the improvement of OA through a mixture of PN and HA in animal experiments. In addition, we examined the relations between pathology, inflammation, and motor activity in the PN and HA mixed group in comparison with the HA only group and PN only group in an animal model of OA.

However, this study had limitations in that, among the various animal models of OA reported to date, we used surgical models limited to ACLT and MMx and did not perform all of the behavioral and pathological tests related to OA. Further studies are needed to examine the role of PN in OA in greater depth, along with an evaluation of the efficacy in OA according to differences in the mixing ratio of HA and PN and the raw material characteristics.

## 4. Materials and Methods

### 4.1. Materials and Manufacturing

A Planetary mixer (BTB solution, Siheung, Republic of Korea, Facility equipment registration number: NFEC-2022-06-279546) was used to mix PN and HA. All products (PNHA/a, PNHA/b, and PN) were manufactured in Joonghun Pharmaceutical’s GMP certified factory.

### 4.2. Animal OA Models

Eight-week-old (300 g) or ten-week-old (350 g) male Sprague–Dawley (SD) rats (Orient Bio, Seong-Nam, Republic of Korea) were used in the experiments. The rats were housed in a temperature-controlled environment (22 °C ± 1 °C, 12 h light/dark cycle; relative humidity, 40–60%) and given ad libitum access to pelleted maintenance diet and drinking water. Animal care and handling were performed in accordance with the guidelines of the Institutional Animal Care and Use Committee of Chungnam National University in Daejeon, Republic of Korea (approval number: CNUH-2022-IA0027). The animals were acclimatized for approximately 2 weeks before the start of the experiment.

For the experiment, 8–12 SD rats were used per group. To create the OA model, each rat was first anesthetized using isoflurane. After shaving and disinfection of the area around the left knee, the knee joint was exposed using the medial parapatellar approach.

The OA model, induced by ACLT and MMx, involved the transection of both the anterior cruciate ligament and patellar tendon [44,58,59]. MMx was performed as previously described [60]. Post-surgery, the joint surface was cleansed with sterile saline and 250 mg/mL ampicillin (Sigma-Aldrich, St. Louis, MO, USA) [61]. The patellar tendon and skin were sutured twice using monofilament 4-0 nylon thread. Sham-operated animals underwent mock surgery, involving an incision in the skin and patellar tendon, followed by rinsing with saline and ampicillin and wound closure. To prevent infection, penicillin was intraperitoneally injected 1–3 days after the surgery.

PNHA/a (Joonghun Pharmaceutical, Seoul, Republic of Korea), PNHA/b (Joonghun Pharmaceutical, Seoul, Republic of Korea), Lufla (Joonghun Pharmaceutical, Seoul, Republic of Korea), and Cureran (Yuhan Corp., Seoul, Republic of Korea) were intra-articularly injected five times, once every week, starting 3 weeks after the surgery. OFT was performed in the 8th week after surgery, followed by the removal of the operated leg. Details of the surgery and sample injection are presented in Figure 1 and Figure 3.

### 4.3. OFT for Locomotor Activity

In animal models, OA manifestations involve a complex interplay between pain and functional impairment. Several biomechanical and functional assessments are available to evaluate the consequences of OA-related defects or disabilities [48]. OFT involves paw-print assessment and locomotion analysis conducted in an open area where rats can freely explore. This method was chosen because distance and speed measurements are indicative of joint function and activity capacity [48,49].

To assess the natural movement activity of the subjects, a chamber measuring 60 cm × 60 cm × 60 cm was utilized. The chamber, constructed from black, high-density, non-porous plastic, was wiped with 70% ethanol before and after each test to eliminate any scent clues from the previous subject rat. Ethanol was allowed to evaporate completely before testing each rat. A digital camera (Canon, Tokyo, Japan), positioned above the chamber and attached to a camera stand, was used to record and evaluate the movements of the rats. The animals were brought from their housing room to the testing room in their home cages and allowed to acclimate for a minimum of 30 min before the test. A single rat was gently removed from its home cage by grasping its tail and placed in the center of the open-field maze. The rat was allowed free and uninterrupted movement within the respective quadrant of the maze for a single 5 min period, during which the camera recorded its movement (The first 2 min were excluded to avoid reflecting emotional states). At the end of the test, the rat was gently picked up, removed from the maze, and returned to its home cage. Fecal pellets were removed, and any spots of urination were cleaned. The maze quadrant’s floor and walls were sprayed with 70% ethanol and wiped with a clean paper towel. The ethanol solution was allowed to dry completely before testing the next subject. The procedure was repeated for each subject. After the experiments, movement patterns were analyzed using the Ethovision XT software version 11 (Noldus, Wageningen, The Netherlands) in the laboratory.

### 4.4. Histological Staining

Following euthanasia, the left knee joints were collected, cleaned of adhering tissues, fixed with 10% neutral buffered formalin (Biosolution, Suwon, Republic of Korea), and decalcified in 10% formic acid or 10% ethylenediaminetetraacetic acid (EDTA) decalcifying solution (Biosolution, Suwon, Republic of Korea) at 4 °C for 4 weeks. Then, the knee joints were dehydrated using ethanol and xylene, embedded in paraffin, and cut longitudinally to obtain slices 6 μm thick. H&E staining and safranin-O Fast Green staining were performed. Safranin O staining was performed using Safranin-O Staining Solution (Sigma-Aldrich, St. Louis, MO, USA), and H&E staining was performed according to the protocol [62]. Cartilage degradation was quantified using the OARSI cartilage grading system (Table 1) [63,64].

### 4.5. Immunofluorescence

Paraffin-embedded sections were subjected to dewaxing and dehydration in an alcohol gradient. For antigen retrieval, the slides were boiled in a citrate buffer (pH 6.0, Sigma Aldrich, St. Louis, MO, USA) for 10 min. After washing with 1X PBS (Biosolution, Suwon, Republic of Korea), the primary antibody diluted in BlockAid™ Blocking Solution (Thermo Fisher Scientific, Waltham, MA, USA) was applied to the slide and allowed to react at 4 °C for more than 16 h. After washing, the secondary antibody, diluted in 1X PBS, was added to the slide and incubated at 37 °C for 60 min. Then, the slides were viewed through a fluorescence microscope ECLIPS Ni (Nikon, Tokyo, Japan) after mounting with Fluoro-mount-G™ Mounting Medium, containing DAPI (Thermo Fisher Scientific, Waltham, MA, USA). Goat anti-rabbit IgG and goat anti-mouse IgG were purchased from Thermo Fisher Scientific, while anti-TNFa and anti-IL6 were obtained from GeneTex (Irvine, CA, USA). Anti-MMP13 and anti-COLII were purchased from AbCam (Cambridge, UK), and IL-1 beta antibody was sourced from Novus biologicals (Centennial, CO, USA). Stained cells were quantified using the Image J (v1.54d) program.

### 4.6. Statistical Analysis

GraphPad Prism Version.8 (GraphPad Software Inc., Boston, MA, USA) was used for the statistical analysis. Data are expressed as means with standard errors of the mean (SEMs). Unpaired *t*-tests were used to compare different groups. A *p*-value < 0.05 was considered statistically significant.

## 5. Conclusions

IA injection of PNHA was found to be an effective treatment for OA. It prevented cartilage matrix degradation and showed anti-inflammatory effects by suppressing TNFα levels. This novel treatment strategy may be an effective alternative for HA products in OA. However, further in-depth research is needed to elucidate the benefits of PNHA compared to PN alone.

## Figures and Tables

**Figure 1 ijms-25-01714-f001:**
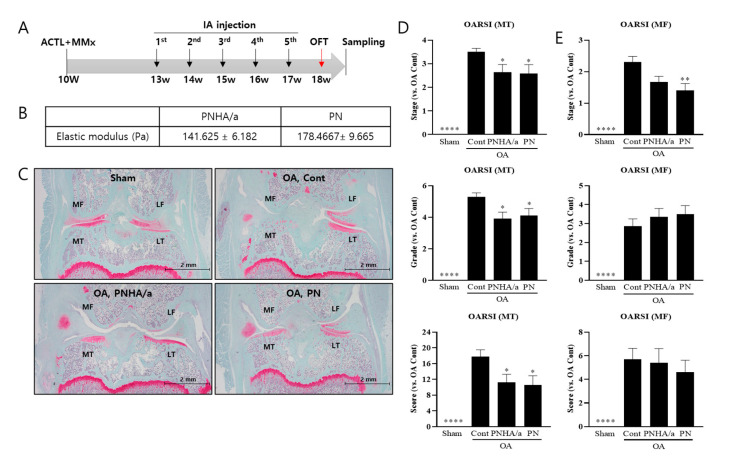
IA injection of PNHA/a delayed cartilage degeneration. (**A**) Experimental sched-ule for surgery and injection. (**B**) Elastic modulus of PNHA/a and PN. (**C**) Safranin O-Fast Green Staining. (**D**,**E**) OARSI score and grade in medial tibia and medial femur. *N* = 10–13 per group. MF, medial femur; LF, lateral femur; MT, medial tibia; LT, lateral tibia; OFT, Open field test; Sham, sham operation; OA, osteoarthritis; Cont, control vehicle; PNHA/a, polynucleotide (2%) with hyaluronic acid (0.5%); PN, polynucleotide (2%). Relative to OA control: * *p* < 0.05, ** *p* < 0.01, **** *p* < 0.0001 (unpaired *t*-test). The data are expressed as the mean ± SEM.

**Figure 2 ijms-25-01714-f002:**
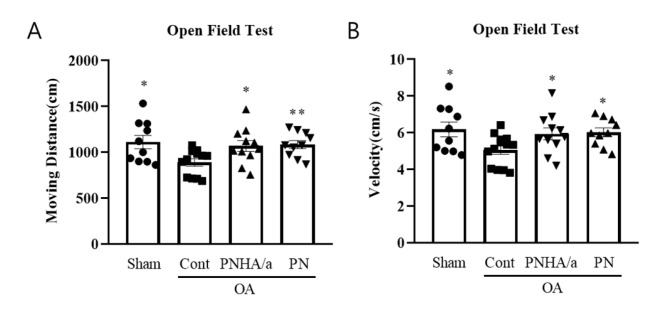
PNHA/a rescued locomotor activity in an OA rat model. (**A**) Moving distance of object from open field test. (**B**) Velocity of object from open field test. *N* = 10–12 per group. Sham, sham operation; OA, osteoarthritis; Cont, control vehicle; PNHA/a, polynucleotide with hyaluronic acid; PN, polynucleotide. Relative to OA control: * *p* < 0.05, ** *p* < 0.01, (unpaired *t*-test). The data are expressed as the mean ± SEM.

**Figure 3 ijms-25-01714-f003:**
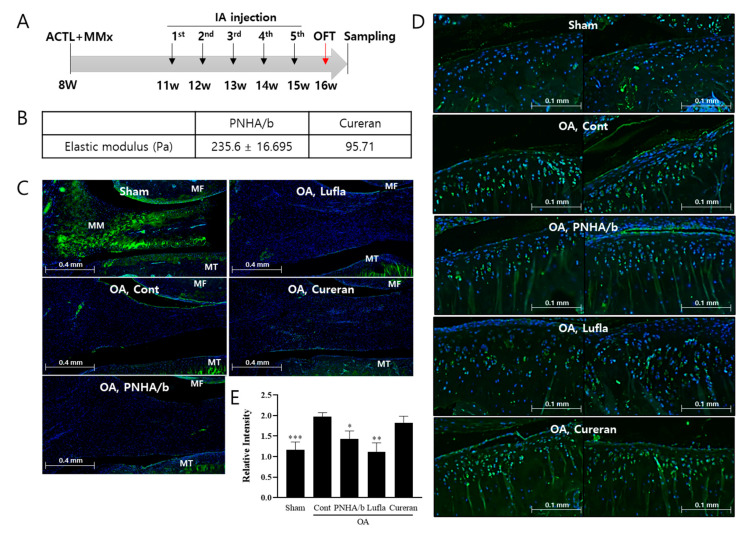
IA injection of PNHA/b suppressed MMP13 expression in cartilage. (**A**) Experimental schedule for surgery and injection. (**B**) Elastic modulus of PNHA/b and Cureran. (**C**) Collagen type 2 immunofluorescence staining of medial joint cartilage. (**D**) MMP13 immunofluorescence staining and quantification of staining amount in medial joint cartilage. (**E**) Graph quantifying MMP13 staining intensity in Figure 1D. *n* = 8–13 per group. MF, medial femur; MT, medial tibia; Sham, sham operation; OA, osteoarthritis; Cont, control vehicle; PNHA/b, polynucleotide (2%) with hyaluronic acid (1%); Lufla, Hyaluronic acid (1%); Cureran, polynucleotide (2%). Relative to OA control: * *p* < 0.05, ** *p* < 0.01, *** *p* < 0.001 (unpaired *t*-test). The data are expressed as the mean ± SEM.

**Figure 4 ijms-25-01714-f004:**
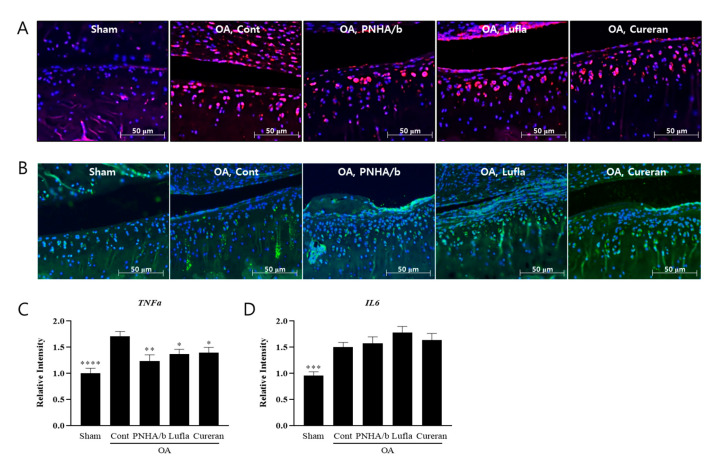
IA injection of PNHA/b reduced TNFα levels in medial tibial cartilage of OA rat model. (**A**) TNFα immunofluorescence staining in medial joint cartilage. (**B**) IL6 immunofluorescence staining. (**C**,**D**) Quantification of TNFα and IL6 staining amount in articular cartilage. *n* = 8–13 per group. Sham, sham operation; OA, osteoarthritis; Cont, control vehicle; PNHA/b, polynucleotide (2%) with hyaluronic acid (1%); Lufla, Hyaluronic acid (1%); Cureran, polynucleotide (2%). Relative to OA control: * *p* < 0.05, ** *p* < 0.01, *** *p* < 0.001, **** *p* < 0.0001, (unpaired *t*-test). The data are expressed as the mean ± SEM.

**Figure 5 ijms-25-01714-f005:**
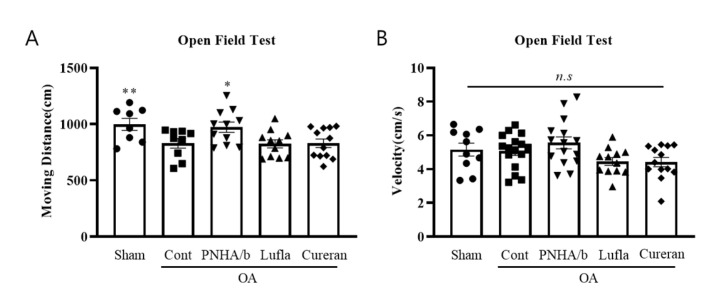
PNHA/b rescued locomotor activity in an OA rat model. (**A**) Moving distance of object from open field test. (**B**) Velocity of object from open field test. *n* = 8–12 per group. Sham, sham operation; OA, osteoarthritis; Cont, control vehicle; PNHA/b, polynucleotide (2%) with hyaluronic acid (0.5%); HA, hyaluronic acid (0.5%). n.s, not significant. Relative to OA control: * *p* < 0.05, ** *p* < 0.01, (unpaired *t*-test). The data are expressed as the mean ± SEM.

**Table 1 ijms-25-01714-t001:** A cartilage histopathology stage and grade assessment grading methodology.

**Stage**	**% Involvement (Surface, Area, Volume)**
Stage 0	No OA activity
Stage 1	<10%
Stage 2	11–25%
Stage 3	26–50%
Stage 4	>50%
**Grade**	**Associated Criteria (Tissue Reaction)**
Grade 0: Surface, cartilage intact	Matrix: normal architecture Cells: intact, appropriate orientation
Grade 1: Surface intact	Matrix: superficial zone intact, oedema and/or superficial fibrillation (abrasion), focal superficial matrix condensation Cells: death, proliferation (clusters), hypertrophy, superficial zone Reaction must be more than superficial fibrillation only
Grade 2: Surface discontinuity	As above +Matrix discontinuity at superficial zone (deep fibrillation) ±Cationic stain matrix depletion (Safranin O or Toluidine Blue) upper 1/3 of cartilage ±Focal peri-chondronal increased stain (mid zone) ±Disorientation of chondron columns Cells: death, proliferation (clusters), hypertrophy
Grade 3: Vertical fissures (clefts)	As above +Matrix vertical fissures into mid zone, branched fissures ±Cationic stain depletion (Safranin O or Toluidine Blue) into lower 2/3 of cartilage (deep zone) ±New collagen formation (polarized light microscopy, Picro Sirius Red stain) Cells: death, regeneration (clusters), hypertrophy, cartilage domains adjacent to fissures
Grade 4: Erosion	Cartilage matrix loss: delamination of superficial layer, mid layer cyst formation Excavation: matrix loss superficial layer and mid zone
Grade 5: Denudation	Sclerotic bone or reparative tissue including fibrocartilage within denuded surface. Microfracture with repair limited to bone surface
Grade 6: Deformation	Bone remodeling (more than osteophyte formation only). Includes: microfracture with fibrocartilaginous and osseous repair extending above the previous surface

## Data Availability

Data contained within the article.

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
