# Peer review of "Effects of a Combination of Polynucleotide and Hyaluronic Acid for Treating Osteoarthritis"

_ijms, 2024, doi:10.3390/ijms25031714_

Round 1
Reviewer 1 Report
Comments and Suggestions for Authors
The article is not of sufficient quality to be published in this journal. Specific questions and points requiring attention are itemized below
1. It is still difficult to find the novelty of the work concerning what has already been published. What is the difference between what is published and what the authors want to publish? It is not clear.
2. Line 42. Explain more about hyaluronic acid. Chemical structure, sources, biological applications, among others.
3. Line 54. "innovative medical products composed of polynucleotides (PNs)" What type???
Reviewer 2 Report
Comments and Suggestions for Authors
The authors combined HA with polynucleotides to treat in OA in a OA model. The manuscript described a novel approach and found some aspects of OA alleviated by this combination.
Abstract and introduction
Line 27: „degenerative disease“ this concept might not be up-to date any longer
Line 38: „Due to these concerns, intermittent or periodic NSAID use is 38 generally recommended.“ Rather Not recommended?
Line 46: „mechanical defense“ sounds strange
Lines 51-52: „However, despite numerous preclinical and clinical reports on its efficacy and safety, the effectiveness of HA in knee OA remains controversial.“ Requires citations reflecting this statement. It is highly contradictory tot he statement in the abstract (line 13): „with well-established efficacy and mechanisms.“
Line 55: „utilized since 2010“ add reference
Lines 66-77: are there rat studies with PN?
Results
Line 82 „by blending 20 mg/mL (2%) PN with/without 5 mg/mL (0.5%) HA“ what ist he rationale of this dosis?
Line 86: „During PNHA/a“ what means „/a“?
Why was only medial evaluated (Fig. 1) indicate which side in the images is medial
Fig. 1: b)elastic modulus: might it be possible to add standard deviation? C) add scale bars
Why do the score values in tibia and femur differ? Why was the treatment effective at the tibia but not femur? Please discuss it more thoroughly
Fig. 2 moving distance% should be explained in the legend, better to label the 3 diagrams A-C?
Fig. 3/4 which cartilage? Tibial or femoral? Please label it if also the menisci are visible
Line 272: transsection of patellar tendon: i do not believe this, might it be the anterior meniscotibial ligament or the medial collateral ligament?
Comments on the Quality of English Languageminor
Reviewer 3 Report
Comments and Suggestions for Authors
Every statements should be supported by specific references. Please check the whole manuscript.
Introduction
The introduction has to be improved in order to better explain osteoarthritis disease.
Osteoarthritis is a disease of the whole joint involving all tissues. Indeed, it is characterized not only by cartilage degeneration, subchondral bone changes, osteophyte formation, and low-grade synovitis but also meniscus degeneration and infrapatellar fat pad inflammation and fibrosis.
Lines 27-29: The role of infrapatellar fat pad is completely lacking. Moreover, menisci degeneration is not mentioned. Please add specific references (DOI:10.3390/biomedicines10061369).
Moreover, all the molecules analysed in this study should be described along with their involvement in the pathology.
Line 34 reference is lacking. Please cite a review e.g. ref 5.
Results
Lines 82-84: please clarify how these concentrations have been chosen.
Line 88: please check supplementary figure 1. It showed data about PNHA/b and no information has been provided about elastic modulus.
Line 106: a specific reference for OARSI score is lacking. Please add.
Lines 117-122: please move this part to the material and methods section.
Lines 160-162: please move this part to the introduction section. Matrix degeneration leads to a cartilage degeneration thus changing the biomechanics of the tissue (DOI:10.3390/pr11041014 and etc.)
Line 162: reference 22 is quite old. Please update.
Lines 168-169: please move this part to the introduction section.
Line 188: please check if figure 6 is correct.
Figures
Please check the supplementary figures. It seemed that supplementary figures 2 and 3 are lacking.
Please improve the quality of the figures, e.g. figure 1D, figure 2, etc.
Please add magnification in the histological figures.
Please check figure 5: moving distance in cm seems the same of moving distance in %.
Discussion
Line 203: please use the abbreviation for osteoarthritis.
Lines 216-217: authors reported one study but two references have been provided. Please check.
Lines 225-227: please add the specific reference of the clinical study authors refer to.
Lines 227-229: please add the specific references of the previous studies authors refer to.
Limits and strengths of the study are lacking. Please add.
Authors should discuss the following systematic review about the topic of the study (doi: 10.1097/MD.0000000000020689).
Materials and methods
Please add the complete information (name, city, country, etc.) about all the materials used.
4.2. Animal OA Models [18, 33]
Line 267: please cite these references in the text at the end of the specific paragraph.
Line 268: Please add rats’ weight, Moreover, describe how rats where housed and in which number, add laboratory conditions (temperature and light–dark-cycle) and which food was provided to the rats.
How many rats were used for the experiments? Please add.
Line 273: please add a specific reference.
Line 302: please add the software’ version.
4.3. Histological Staining
Please add the complete information (name, city, country, etc.) about all the materials used.
Line 318: please provide the complete information about the fluorescence microscope.
Line 323: please add the software’ version.
Table 1 is lacking. Please clarify.
Round 2
Reviewer 1 Report
Comments and Suggestions for Authors
The article can be accepted
Author Response
Thank you very much for accepting my thesis.
Reviewer 2 Report
Comments and Suggestions for Authors
My previous comments have been adressed, the manuscript was sufficiently revised.
line 186-89: "sched-ule" and " immunofluores-cence" hyphen could be removed. Fig. 3: expression... "in cartilage" could be added in the heading of the legend
line 198: "Data not shown" could it be shown as supplemental figure.
There is a surplus point in line 202.
line 207: "in medial tibia" better to write in medial tibial cartilage?
Author Response
Thank you for your positive consideration of our paper. Please check the rebuttal letter as it is attached as a file.

Reviewer 3 Report
Comments and Suggestions for Authors
No comments. Thank you
Comments on the Quality of English LanguageNo comments. Thank you
Author Response

(The authors gave the same response as above.)
